# Analysis of Fear Post COVID in First-Year Students after the Incorporation to the Classroom: Descriptive Study in University Students of Health Sciences

**DOI:** 10.3390/healthcare9121621

**Published:** 2021-11-24

**Authors:** Pablo A. Cantero-Garlito, Marta Rodríguez-Hernández, Esther Moraleda-Sepúlveda, Begoña Polonio-López, Félix Marcos-Tejedor

**Affiliations:** 1Department of Nursing, Physiotherapy and Occupational Therapy, Faculty of Health Sciences, University of Castilla-La Mancha, 45600 Talavera de la Reina, Spain; Pablo.Cantero@uclm.es (P.A.C.-G.); Marta.RHernandez@uclm.es (M.R.-H.); Begona.polonio@uclm.es (B.P.-L.); 2Department of Psychology, Faculty of Health Sciences, University of Castilla-La Mancha, 45600 Talavera de la Reina, Spain; 3Department of Medical Sciences, Faculty of Health Sciences, University of Castilla-La Mancha, 45600 Talavera de la Reina, Spain; Felix.MarcosTejedor@uclm.es

**Keywords:** COVID-19 pandemic, university students, fear

## Abstract

Background: After the onset of the COVID-19 pandemic, social restriction measures were implemented, among them, the adaptation of university teaching to online modality until the end of the 2019–2020 school year in order to stop the spread of the SARS-CoV-2 virus. At the beginning of the 2020–2021 school year, the Spanish universities opted for face-to-face teaching. To that end, different special measures and adaptations were implemented in higher education facilities, aimed at minimizing the risk of infection and ensuring safe face-to-face learning. The objective was to explore and describe the level of fear of first-year students after the start of in-person classes in the context of the COVID-19 pandemic. Methods: The sample was 185 first-year students who were evaluated on the first day of class. For that purpose, an ad-hoc questionnaire was administered to collect demographic information and to find the level of fear and concern. The Fear of COVID-19 Scale was used to assess the severity of the participants’ fear of the pandemic situation. Results: The results indicate that participating university population does not report fear of the virus, but they describe various psychosomatic characteristics, such as increased pulse rate and heart palpitations (*p* = 0.008) and insomnia (*p* = 0.05) when they think about infection with coronavirus. Nevertheless, when data are disaggregated by gender, we observe differences specifically in women (83.2%), such as fear (*p* = 0.006) and sweaty hands when they think of the virus (*p* = 0.023). Conclusions: Incoming university freshmen do not express concern or fear of potential infection with COVID-19, but they are concerned about family transmission after beginning face-to-face classes.

## 1. Introduction

The emergence of the SARS-CoV-2 virus in December 2019 in China led the World Health Organization (WHO) to declare the COVID-19 pandemic an international public health emergency in March 2020. This has had a very significant impact on people’s daily lives, and has had important consequences on public health, economy, employment, and environment [1]. As a result, the Spanish Government declared the state of emergency in Spain on 14 March 2020. One of the immediate implications of this was the closing of all education institutions, including universities.

In order to continue with university teaching, the Ministry of Universities produced a series of reports for the adaptation of the Spanish university system from in-person to online modality in the context of COVID-19, including reflections on general criteria for the adaptation of the Spanish university system in response to the COVID-19 pandemic, during the 2019–2020 school year (14 April 2020), and a report on university online assessment initiatives and tools in the context of COVID-19 (8 April). Likewise, the Conference of Rectors of Spanish Universities (CRUE) prepared the report on distance assessment procedures, a study on the impact of its implementation in Spanish universities and recommendations. Following this line, each university proposed the guidelines for the adoption of the distance modality until the end of the school year. Thus, the classes of the 2019–2020 year in educational centers finished in a primarily online format.

Nevertheless, the evolution of the pandemic led to a different scenario for the year 2020–2021, and it was resolved that students returned to the classrooms. In the case of universities, the relevant ministry made a set of recommendations to the university community for the modification of the new school year based mainly on an adapted face-to-face format, which lays the basis for an in-person return to the classroom with the maximum safety measures [2]. Subsequently, in most cases there has been a return to a face-to-face modality, sometimes combined with a hybrid model (combining teaching tools and resources from the in-person and the distance modalities).

Different studies conducted have highlighted the psychological effects and implications of this pandemic on the general population, such as anxiety, the fear or how demographics and socioeconomic status conditions can interfere with the psychological health of students during the interruption of the face-to-face teaching activities [3,4,5,6,7].

One of the most important studies carried out so far in university population is Martínez-Lorca et al [8], which evaluated 606 students from the Universidad de Castilla-La Mancha (UCLM) with the Spanish version and validation of the tool The Fear of COVID-19 Scale (FCV-19S) [9]. These authors point out that university students experience less fear (medium-high levels) compared to general population [9,10]. In addition, this population group sometimes presents psychological characteristics associated with lower risk perception and greater sense of control [8]. These moderate results make young university students be considered more likely not to comply with the confinement measures proposed by the institutions and to be more relaxed, which directly implies a lower compliance with recommended health behaviors [11].

These authors also suggest that one of the factors that influences the perception of fear in university students the most is the year they are enrolled in [7]. The youngest students (first year) present a higher level of fear than the students in later years, suggesting that age is an important indicator and predictor of the level of fear. However, age-dependent results are not consistent with other studies conducted in other countries (for example, Iran, Bangladesh, and Italy), where other research does not find significant differences between age and the level of fear [9,10,12]. 

On the other hand, there are studies that have analyzed how the fear of COVID can affect the health status of the population, and specifically how it can affect the return to face-to-face teaching activities of teachers and, especially of those with chronic pathology [13,14]. Nevertheless, there are not many studies yet that analyze the aspects related to the return to the classroom of university students.

The objective of this study is to explore and describe the level of fear of university first-year students after the start of the face-to-face teaching activities in the context of the COVID-19 pandemic. 

## 2. Materials and Methods

### 2.1. Participants

The study was conducted in the Faculty of Health Sciences in the Universidad de Castilla-La Mancha (from now on UCLM), located in Talavera de la Reina. The reference population is defined by the groups of students enrolled in the first year of the bachelor’s degree of occupational therapy, speech therapy, nursing, podiatry, and double degree in nursing and podiatry. The sample was composed of 185 students from the different degrees (mean age = 20.3, SD = 4.6). The characteristics of the sample are shown in Table 1.

### 2.2. Procedure

For this research, the potential participants were informed verbally about the characteristics of the study during the first day of class of the new groups (first year of the degree) in a face-to-face meeting. Once informed, the students filled in an informed consent form in order to accept their participation. All the participants gave informed consent, with due regard for the Declaration of Helsinki. Finally, the Clinical Research Ethics Committee of the Integrated Area of Talavera de la Reina 49/2020 has issued a favorable opinion.

### 2.3. Instrument

In order to find out about the characteristics of the sample, an ad-hoc questionnaire was first administered to collect demographic information (sex, age and degree) and to know the level of fear and concern about the beginning of in-person classes in the faculty. The students then completed The Fear of COVID-19 Scale, which was developed in 2020 by Perz, C.A et al. [4] and later validated in Spain by Martínez-Lorca, M et al. in that same year [8]. This instrument was created to measure the severity of people’s fear of COVID-19 using a scale of five Likert-type items (1 to 5 points). The responses include: “strongly disagree”, “disagree”, “neither agree nor disagree”, “agree”, and “strongly agree”. The minimum score for each item is 1, and the maximum is 5.

### 2.4. Analysis

Once the fieldwork was completed, all the information gathered was processed, in order to systematize it and make it comprehensible. Of the various existing methods for this, and taking into account the size of the sample, the number of questions in the questionnaire and in the scale used and the available resources, we opted for manual processing. Data sheets were used with the data collected in order to analyze the results. It was decided to present here the information obtained during the fieldwork by means of tables, which show means and the percentage of the response frequency for each of the variables. In addition, it was examined whether there were significant differences (*p* values) by sex (men vs. women), age and degree. Descriptive statistics were used for the variables: absolute frequencies, means and percentages. The results were expressed in percentages and 95% confidence intervals. The significance level was set at 0.05. The Chi-squared test was used for comparisons. The data were analyzed with the statistical package IBM SPSS Statistics (version 22.0 SPSS for Windows; SPSS Inc., Chicago, IL., USA).

## 3. Results

### 3.1. The Fear of COVID-19 Scale 

The results of the questionnaire were obtained from the analysis of the variables included in The Fear of COVID-19 Scale. The findings indicate that there is not great fear of the virus among university freshman population in the selected campus. The mean score is 17.53 (SD = 5.56). However, we find differences in behavior according to sex and age group. Women present a higher percentage of fear of coronavirus (*n* = 65; %: 42.2; *p* = 0.006) and of indifference to the feeling of sweaty hands when they think about the virus (*n* = 22; %: 14.3; *p* = 0.023) (Table 2). 

In the youngest age group (under 20 years old), we find increased pulse and palpitations when they think about infection with coronavirus (*n* = 18; %: 16.3; *p* = 0.008). Nonetheless, the oldest students (20 or older) are the ones who show most insomnia due to concern about infection (*n* = 4; % = 6.0, *n* = 1; % = 12.5; *p* = 0.050) (Table 3).

### 3.2. Assessment of Fear and Concern over Infection and/or Isolation 

In the questionnaire administered to find out the level of fear and concern about infection with COVID-19 and in-person attendance to classes in the campus, higher response rates are observed in the categories “agree” and “strongly agree”. Moreover, 34.6% of the sample say that they are afraid of infection at the university and 49.7% of the respondents show concern about infecting their families due to their attendance of classes. Only 11.3% would have preferred the classes of their degree to have been online. Finally, 41.6% of students show concern about a possible lockdown outside their city of origin (Table 4). 

## 4. Discussion

The rapid spread of the COVID-19 pandemic in Spain in March 2020 prompted the university authorities to make the decision to cancel classes which, until then, had been in-person, and to opt for a model in which both classes and the different assessment tests were online. These organizational decisions were extended until the end of the year 2019–2020.

With the start of the new school year, the measures implemented in the universities over the following months (distance between students, more open spaces, reduced capacities, weekly rotation of students in the classrooms) made it possible for many students to join in-person classes for the first time. It was especially relevant to analyze whether these students, who very often feel anxiety and fear upon their arrival at university [2], experienced levels of fear of COVID-19 that may influence their academic performance or complicate their process of adaptation to the new educational environment and their relationship with their peers.

However, in this survey, students do not show particularly relevant levels of fear of COVID-19. Unlike the study by Perz [4] where no difference was found by sex of students as regards levels of fear of COVID, we do observe significant differences. This is consistent with Tzur Bitan [13], who found that gender, socio-demographic status, chronic diseases, belonging to risk groups and the fact that a family member died from COVID-19 were all factors positively associated with fear of the virus. 

Likewise, students feel more inclined to receive their university training in a face-to-face manner, something that is in line with other studies that underscore the fear of students to receive worse instruction or simply to “lose the year” because attendance-based university centers use online teaching-learning methodologies [5].

As can be seen, the results are discussed with work carried out during confinement, hence there are these discrepancies in the data obtained, which is possibly due to the acceptance of the evolution of the pandemic.

Even so, as Cao [5] points out, the mental health of university students should be monitored during epidemics, given that, in addition, it has become evident that fear, as a multifaceted factor, can be one of the most important underlying elements leading to deterioration of mental health and welfare [15]. The existing variability in the epidemiological situation, as regards levels of infection, in measures to prevent the spread of the virus in university environments, the impact of measures on the economic situation of families can lead to a deterioration of mental health, with a resulting effect on well-being and academic performance may also be highlighted.

## 5. Conclusions

On the whole, university first-year students who begin their training in an in-person modality do not experience great fear of COVID-19, although they present increased pulse and palpitations and insomnia as common and emergent elements after the pandemic. In the female population, a higher and statistically significant level of fear is observed. Nevertheless, it is worth noting the proportion of students that express fear of infection at the university and, even more so, of the possibility to infect the people they live with or their families.

There is a need for a larger number of studies which examine how fear of COVID affects the academic performance of freshman to senior university students.

## 6. Limitations

As possible limitations of the study, we can highlight that the analysis of the results of the work is based on a research technique on a group studied that is not very large.

## Figures and Tables

**Table 1 healthcare-09-01621-t001:** Socio-demographic characteristics of the sample (*n* = 185).

Socio-Demographic Characteristics	*n*	%
Age		
Mean (SD) (range)	185	20.3 (4.6) (18–50)
Under age 20	110	59.5
Age 20 to 25	67	36.2
Over age 25	8	4.3
Sex		
Man	31	16.8
Woman	154	83.2
Degree		
Speech therapy	53	28.6
Occupational therapy	64	34.6
Podiatry	26	14.1
Double degree nursing + podiatry	6	3.2
Nursing	36	19.5

**Table 2 healthcare-09-01621-t002:** Differences by sex in The Fear of COVID-19 Scale (*n* = 185).

The Fear of COVID-19 Scale	Man	Woman	*p*
	*n* (%) (95% CI) *	*n* (%) (95% CI)	
**Q1. I am very scared of coronavirus-19.**			* **0.006** *
Strongly disagree	2 (6.5) (1.4–19.1)	4 (2.6) (0.9–6.1)	
Disagree	10 (32.3) (17.9–49.7)	19 (12.3) (7.9–18.2)	
Neither agree nor disagree	4 (12.9) (4.5–27.8)	66 (42.9) (35.2–50.7)	
Agree	12 (38.7) (23.2–56.2)	46 (29.9) (23.1–37.4)	
Strongly agree	3 (9.7) (2.8–23.6)	19 (12.3) (7.9–18.2)	
**Q2. Thinking about coronavirus-19 makes me feel uncomfortable.**			*0.519*
Strongly disagree	2 (6.5) (1.4–19.1)	12 (7.8) (4.3–12.8)	
Disagree	10 (32.3) (17.9–49.7)	32 (20.8) (15.0–27.7)	
Neither agree nor disagree	7 (22.6) (10.7–39.3)	54 (35.1) (27.9–42.8)	
Agree	7 (22.6) (10.7–39.3)	38 (24.7) (18.4–31.9)	
Strongly agree	5 (16.1) (6.4–31.8)	18 (11.7) (7.3–17.5)	
**Q3. My hands get sweaty when I think about coronavirus-19.**			* **0.023** *
Strongly disagree	18 (58.1) (40.6–74.1)	98 (63.6) (55.8–70.9)	
Disagree	8 (25.8) (13.0–42.9)	30 (19.5) (13.8–26.3)	
Neither agree nor disagree	1 (3.2) (0.4–14.1)	22 (14.3) (9.4–20.5)	
Agree	4 (12.9) (4.5–27.8)	4 (2.6) (0.9–6.1)	
Strongly agree	0	0	
**Q4. I am scared of dying from coronavirus-19.**			*0.552*
Strongly disagree	9 (29.0) (15.4–46.3)	27 (17.5) (12.2–24.1)	
Disagree	8 (25.8) (13.0–42.9)	36 (23.4) (17.2–30.5)	
Neither agree nor disagree	4 (12.9) (4.5–27.8)	33 (21.4) (15.5–28.4)	
Agree	6 (19.4) (8.5–35.6)	32 (20.8) (15.0–27.7)	
Strongly agree	4 (12.9) (4.5–27.8)	26 (16.9) (11.6–23.4)	
**Q5. When I see news and stories about coronavirus-19 in social media, I feel nervous or anxious.**			*0.61*
Strongly disagree	5 (16.1) (6.4–31.8)	19 (12.3) (7.9–18.2)	
Disagree	7 (22.6) (10.7–39.3)	40 (26.0) (19.5–33.3)	
Neither agree nor disagree	12 (38.7) (23.2–56.2)	45 (29.2) (22.5–36.7)	
Agree	3 (9.7) (2.8–23.6)	31 (20.1) (14.4–27.0)	
Strongly agree	4 (12.9) (4.5–27.8)	19 (12.3) (7.9–18.2)	
**Q6. I can’t sleep because I am worried about catching coronavirus-19.**			*0.552*
Strongly disagree	21 (67.7) (50.3–82.1)	97 (63.0) (55.2–70.3)	
Disagree	7 (22.6) (10.7–39.3)	37 (24.0) (17.8–31.2)	
Neither agree nor disagree	2 (6.5) (1.4–19.1)	13 (8.4) (4.8–13.6)	
Agree	0	6 (3.9) (1.6–7.9)	
Strongly agree	1 (3.2) (0.4–14.1)	1 (0.6) (0.1–3.0)	
**Q7. My heart beats faster when I think about catching coronavirus-19.**			*0.82*
Strongly disagree	13 (41.9) (25.9–59.4)	56 (36.4) (29.1–44.2)	
Disagree	10 (32.3) (17.9–49.7)	41 (26.6) (20.1–34.0)	
Neither agree nor disagree	4 (12.9) (4.5–27.8)	31 (20.1) (14.4–27.0)	
Agree	3 (9.7) (2.8–23.6)	21 (13.6) (8.9–19.7)	
Strongly agree	1 (3.2) (0.4–14.1)	5 (3.2) (1.2–7.0)	

* 95% CI: 95% confidence interval; bold: Statistically significant difference.

**Table 3 healthcare-09-01621-t003:** Differences by age in The Fear of COVID-19 Scale (*n* = 185).

The Fear of COVID–19 Scale	Under Age 20	Age 20 to 25	Over Age 25	*p*
	*n* (%) (95% CI) *	*n* (%) (95% CI)	*n* (%) (95% CI)	
**Q1. I am very scared of coronavirus-19.**				*0.697*
Strongly disagree	3 (2.7) (0.8–7.1)	3 (4.5) (1.3–11.5)	0	
Disagree	16 (14.5) (8.9–22.0)	12 (17.9) (10.2–28.3)	1 (12.5) (1.4–45.4)	
Neither agree nor disagree	38 (34.5) (26.4–43.7)	28 (41.8) (30.5–53.7)	4 (50.0) (19.9–80.1)	
Agree	39 (35.5) (27.0–44.7)	18 (26.9) (17.4–38.3)	1 (12.5) (1.4–45.4)	
Strongly agree	14 (12.7) (7.5–19.9)	6 (9.0) (3.8–17.5)	2 (25.0) (5.6–59.2)	
**Q2. Thinking about coronavirus-19 makes me feel uncomfortable.**				*0.356*
Strongly disagree	9 (8.2) (4.1–14.4)	4 (6.0) (2.0–13.6)	1 (12.5) (1.4–45.4)	
Disagree	26 (23.6) (16.4–32.2)	14 (20.9) (12.5–31.7)	2 (25.0) (5.6–59.2)	
Neither agree nor disagree	38 (34.5) (26.2–43.7)	22 (32.8) (22.5–44.6)	1 (12.5) (1.4–45.4)	
Agree	23 (20.9) (14.1–29.2)	21 (31.3) (21.2–43.1)	1 (12.5) (1.4–45.4)	
Strongly agree	14 (12.7) (7.5–19.9)	6 (9.0) (3.8–17.5)	3 (37.5) (11.9–70.5)	
**Q3. My hands get sweaty when I think about coronavirus-19.**				*0.322*
Strongly disagree	69 (62.7) (53.5–71.3)	44 (65.7) (53.8–76.2)	3 (37.5) (11.9–70.5)	
Disagree	23 (20.9) (14.1–29.2)	14 (20.9) (12.5–31.7)	1 (12.5) (1.4–45.4)	
Neither agree nor disagree	13 (11.8) (6.8–18.8)	7 (10.4) (4.8–19.4)	3 (37.5) (11.9–70.5)	
Agree	5 (4.5) (1.8–9.7)	2 (3.0) (0.6–9.2)	1 (12.5) (1.4–45.4)	
Strongly agree	0	0	0	
**Q4. I am scared of dying from coronavirus-19.**				*0.703*
Strongly disagree	21 (19.1) (12.6–27.2)	12 (17.9) (10.2–28.3)	3 (37.5) (11.9–70.5)	
Disagree	27 (24.5) (17.2–33.2)	15 (22.4) (13.7–33.4)	2 (25.0) (5.6–59.2)	
Neither agree nor disagree	20 (18.2) (11.8–26.2)	16 (23.9) (14.9–35.0)	1 (12.5) (1.4–45.4)	
Agree	26 (23.6) (16.4–32.2)	12 (17.9) (10.2–28.3)	0	
Strongly agree	16 (14.5) (8.9–22.0)	12 (17.9) (10.2–28.3)	2 (25.0) (5.6–59.2)	
**Q5. When I see news and stories about coronavirus-19 in social media, I feel nervous or anxious.**				*0.344*
Strongly disagree	11 (10.0) (5.4–16.6)	13 (19.4) (11.3–30.0)	0	
Disagree	27 (24.5) (17.2–33.2)	19 (28.4) (18.6–39.9)	1 (12.5) (1.4–45.4)	
Neither agree nor disagree	35 (31.8) (23.7–40.9)	17 (25.4) (16.1–36.7)	5 (62.5) (29.5–88.1)	
Agree	23 (20.9) (14.1–29.2)	10 (14.9) (7.9–24.7)	1 (12.5) (1.4–45.4)	
Strongly agree	14 (12.7) (7.5–19.9)	8 (11.9) (5.8–21.3)	1 (12.5) (1.4–45.4)	
**Q6. I can’t sleep because I am worried about catching coronavirus-19.**				* **0.05** *
Strongly disagree	71 (64.5) (55.3–73.0)	43 (64.2) (52.3–74.9)	4 (50.0) (19.9–80.1)	
Disagree	27 (24.5) (17.2–33.2)	16 (23.9) (14.9–35.0)	1 (12.5) (1.4–45.4)	
Neither agree nor disagree	9 (8.2) (4.1–14.4)	4 (6.0) (2.0–13.6)	2 (25.0) (5.6–59.2)	
Agree	3 (2.7) (0.8–7.1)	3 (4.5) (1.3–11-5)	0	
Strongly agree	0	1 (1.5) (0.2–6.8)	1 (12.5) (1.4–45.4)	
**Q7. My heart beats faster when I think about catching coronavirus-19.**				* **0.008** *
Strongly disagree	39 (35.5) (27.0–44.7)	28 (41.8) (30.5–53.7)	2 (25.0) (5.6–59.2)	
Disagree	28 (25.5) (18.0–34.2)	22 (32.8) (22.5–44.6)	1 (12.5) (1.4–45.4)	
Neither agree nor disagree	25 (22.7) (15.7–31.2)	7 (10.4) (4.8–19.4)	3 (37.5) (11.9–70.5)	
Agree	16 (14.5) (8.9–22.0)	8 (11.9) (5.8–21.3)	0	
Strongly agree	2 (1.8) (0.4–5.7)	2 (3.0) (0.6–9.2)	2 (25.0) (5.6–59.2)	

* 95% CI: 95% confidence interval; bold: Statistically significant difference.

**Table 4 healthcare-09-01621-t004:** Assessment of fear and concern over infection and/ or isolation (*n* = 185).

Questions	1	2	3	4	5
**Q1**. I am scared of getting infected at the University.	12 (6.5) (3.6–10.7)	23 (12.4) (8.3–17.8)	59 (31.9) (25.5–38.8)	59 (17.3) (25.5–38.8)	32 (17.3) (12.4–23.2)
**Q2.** I am worried that there is a lockdown and I have to stay in Talavera.	31 (16.8) (11.9–22.6)	15 (8.1) (4.8–12.7)	30 (16.2) (11.4–22.0)	32 (17.3) (12.4–23.2)	77 (41.6) (34.7–48.8)
**Q3.** I fear how the pandemic may affect my university studies.	1 (0.5) (0.1–2.5)	7 (3.8) (1.7–7.3)	18 (9.7) (6.1–14.6)	71 (38.4) (31.6–45.5)	88 (47.6) (40.5–54.8)
**Q4.** I worry that I may infect my family by attending university.	4 (2.2) (0.7–5.1)	11 (5.9) (3.2–10.1)	19 (10.3) (6.5–15.3)	59 (31.9) (25.5–38.8)	92 (49.7) (42.6–56.9)
**Q5****.** I am scared of being isolated and not following classes normally.	5 (2.7) (1.0–5.8)	4 (2.2) (0-7–5.1)	27 (14.6) (10.1–20.2)	63 (34.1) (27.5–41.1)	86 (46.5) (39.4–53.7)
**Q6.** I would have preferred classes to be online.	99 (53.5) (46.3–60.6)	32 (17.3) (12.4–23.2)	33 (17.8) (12.8–23.8)	11 (5.9) (3.2–10.1)	10 (5.4) (2.8–9.4)

Response categories: 1 = strongly disagree; 2 = disagree; 3 = neither agree nor disagree; 4 = agree and 5 = strongly agree. The table shows *n* (%) (95% CI). CI: confidence interval.

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
