# Peer review of "Analysis of Fear Post COVID in First-Year Students after the Incorporation to the Classroom: Descriptive Study in University Students of Health Sciences"

_healthcare, 2021, doi:10.3390/healthcare9121621_

Round 1
Reviewer 1 Report
- The authors write in the abstract that the 2020/2021 academic year has begun in a face-to-face mode in Spanish universities, when in some of them this has not been the case, opting for a semi-face-to-face mode.
- In the introduction the authors put a website. They have to cite the information correctly, not put the link.
- Methodology: How was the questionnaire completed? Paper-based? Online? Were they all valid? Specify this process
- The data analysis section is very poor statistically speaking. Was the normality test done? Were parametric or non-parametric tests used? What statistical software was used?
- In the results section, tables should be smaller in size.
- The last table (4) is not well understood: Is it a likert scale from 1 to 5? But what does 1 and 5 mean? Is 1 a little agree or a lot agree or what? This table is not commented either.
- There is talk of palpitations, increased pulse, insomnia... as symptoms that can result from fear of COVID, but are these symptoms measured in any way? Or only through the questionnaire?
- The discussion section is very brief. It is recommended that the authors expand it by focusing more, for example, on gender differences.
- Differentiate the conclusions section from the discussion section
I think it is a very interesting and necessary article, because it is necessary to know how our students feel when they attend class.
I recommend the authors to fix what I have said to improve the quality of the manuscript.
Author Response
The authors write in the abstract that the 2020/2021 academic year has begun in a face-to-face mode in Spanish universities, when in some of them this has not been the case, opting for a semi-face-to-face mode.
- In the introduction the authors put a website. They have to cite the information correctly, not put the link.
This link has been added.
- Methodology: How was the questionnaire completed? Paper-based? Online? Were they all valid? Specify this process
Yes, the questionnaire was completed by paper form and was filled by the students. All students filled completely the Fear of COVID-19 Scale.
- The data analysis section is very poor statistically speaking. Was the normality test done? Were parametric or non-parametric tests used? What statistical software was used?
It has been explained in the analysis part
- In the results section, tables should be smaller in size.
Now, they are smaller.
- The last table (4) is not well understood: Is it a likert scale from 1 to 5? But what does 1 and 5 mean? Is 1 a little agree or a lot agree or what? This table is not commented either.
The meaning about Response categories are: 1= strongly disagree; 2= disagree; 3= neither agree nor disagree; 4= agree; and 5= strongly agree.
- There is talk of palpitations, increased pulse, insomnia... as symptoms that can result from fear of COVID, but are these symptoms measured in any way? Or only through the questionnaire?
These symptoms haven´t been measured in any way. It consist on the answers of the questionnaire.
- The discussion section is very brief. It is recommended that the authors expand it by focusing more, for example, on gender differences.
This part has been changed.
- Differentiate the conclusions section from the discussion section
This part has been changed.
Reviewer 2 Report
The authors, in the article "An exploration of fear in incoming university first-year students in the context of the COVID-19 pandemic" present a study conducted on the fear of covid-19 experienced by first-year university students.
While it is very important to study what people are feeling about the pandemic, the proposed study provides very little evidence for increasing scientific knowledge.
The literature review, as can also be seen from the number of bibliographic references, is practically non-existent. In addition, there are also dubious bibliographic references such as the number 7.
The study involves 185 students who are given the fear of covid-19 scale alone, in addition to socio-demographic information.
The use of this scale alone does not allow to grasp the links between this construct and other constructs or behaviors.
The analyzes for calculating the differences between groups are presented in a non-academic way.
The hypotheses and the comments on the results were absent.
For these reasons, the article cannot be accepted for publication in a highly scientific journal such as Health Care.
Author Response
Reviewer 2
The literature review, as can also be seen from the number of bibliographic references, is practically non-existent. In addition, there are also dubious bibliographic references such as the number 7.
More bibliography has been added.
The study involves 185 students who are given the fear of covid-19 scale alone, in addition to socio-demographic information.
The use of this scale alone does not allow to grasp the links between this construct and other constructs or behaviors.
We haved followed the ítems of the Fear of COVID-19 Scale
The analyzes for calculating the differences between groups are presented in a non-academic way.
This part has been changed.
The hypotheses and the comments on the results were absent.
This part has been changed.
Reviewer 3 Report
The COVID-19 pandemic is undoubtedly a very traumatic event that has had (and still has) a great impact on the functioning of many institutions, including universities. Each article on the psychosocial functioning of young people in these difficult times is interesting and broadens our view about potential risk factors for their development and career. The presented work concerns a very important problem, which is the students' fear of the negative consequences of taking up face-to-face education during a pandemic.
- Is it possible to provide the date and number of the consent of the Ethics Committee for the conducted research?
- I have some doubts whether people over the age of 25 (and there is a person aged 50) should participate in the study. Such people have completely different life experiences compared to younger students.
- There is a litle bit sex-bias in sample grup. Is this connected with gender structure of students at the university?
- The tables require editing - the data in the lines do not match the questions of the questionnaire
- Major question: Is it possible to perform analogous statistical analyzes in table 4 as in the data from tables 2 (by sex) and 3 (by age)? This will make it possible to unform the analyzes and discussion of results.
Author Response
Is it possible to provide the date and number of the consent of the Ethics Committee for the conducted research?
This number has been added.
I have some doubts whether people over the age of 25 (and there is a person aged 50) should participate in the study. Such people have completely different life experiences compared to younger students.
Yes, we agree but all people are first year student and its characteristics could be similar in order to go to university.
There is a litle bit sex-bias in sample grup. Is this connected with gender structure of students at the university?
We aren´t sure about there is a Little big sex-bias. Our students are majority in females.
The tables require editing - the data in the lines do not match the questions of the questionnaire
These tables have been modified.
Major question: Is it possible to perform analogous statistical analyzes in table 4 as in the data from tables 2 (by sex) and 3 (by age)? This will make it possible to unform the analyzes and discussion of results.
We have improved the tables
Round 2
Reviewer 1 Report
The corrections made are satisfactory for me. However, I would suggest adding a part: Limitations - with the limitations of the study, for example: the studied group is not very large, the analysis is based on one research technique.
Author Response
We have included a paragraph about limitations of the study.
Thank you for your comments

Reviewer 2 Report
I would like to congratulate the authors for their work. They have followed the suggested recommendations and have undoubtedly improved the quality of the manuscript.
Author Response
Thank you for your comments.
We have improved the English language and style

Reviewer 3 Report
I would recommend that the authors expand on their description of the Fear of COVID-19 Scale by describing how the instrument was developed, by whom, and psychometric properties of the scale. Since the entire results of the study hinge on this scale, which is a new one, more information about the scale needs to be provided, especially about its psychometric soundness.
Author Response
We have included a new sentence about the Fear of COVID-19 Scale
Thank your for your comments
